# Exact Learning Augmented Naive Bayes Classifier

**DOI:** 10.3390/e23121703

**Published:** 2021-12-20

**Authors:** Shouta Sugahara, Maomi Ueno

**Affiliations:** Graduate School of Informatics and Engineering, The University of Electro-Communications, 1-5-1, Chofugaoka, Chofu-shi, Tokyo 182-8585, Japan; ueno@ai.lab.uec.ac.jp

**Keywords:** augmented naive Bayes classifier, Bayesian networks, classification, structure learning

## Abstract

Earlier studies have shown that classification accuracies of Bayesian networks (BNs) obtained by maximizing the conditional log likelihood (CLL) of a class variable, given the feature variables, were higher than those obtained by maximizing the marginal likelihood (ML). However, differences between the performances of the two scores in the earlier studies may be attributed to the fact that they used approximate learning algorithms, not exact ones. This paper compares the classification accuracies of BNs with approximate learning using CLL to those with exact learning using ML. The results demonstrate that the classification accuracies of BNs obtained by maximizing the ML are higher than those obtained by maximizing the CLL for large data. However, the results also demonstrate that the classification accuracies of exact learning BNs using the ML are much worse than those of other methods when the sample size is small and the class variable has numerous parents. To resolve the problem, we propose an exact learning augmented naive Bayes classifier (ANB), which ensures a class variable with no parents. The proposed method is guaranteed to asymptotically estimate the identical class posterior to that of the exactly learned BN. Comparison experiments demonstrated the superior performance of the proposed method.

## 1. Introduction

Classification contributes to solving real-world problems. The naive Bayes classifier, in which the feature variables are conditionally independent given a class variable, is a popular classifier [1]. Initially, the naive Bayes was not expected to provide highly accurate classification, because actual data were generated from more complex systems. Therefore, the general Bayesian network (GBN) with learning by marginal likelihood (ML) as a generative model was expected to outperform the naive Bayes, because the GBN is more expressive than the naive Bayes. However, Friedman et al. [2] demonstrated that the naive Bayes sometimes outperformed the GBN using a greedy search to find the smallest minimum description length (MDL) score, which was originally intended to approximate ML. They explained the inferior performance of the MDL by decomposing it into the log likelihood (LL) term, which reflects the model fitting to training data, and the penalty term, which reflects the model complexity. Moreover, they decomposed the LL term into a conditional log likelihood (CLL) of the class variable given the feature variables, which is directly related to the classification, and a joint LL of the feature variables, which is not directly related to the classification. Furthermore, they proposed conditional MDL (CMDL), a modified MDL replacing the LL with the CLL.

Consequently, Grossman and Domingos [3] claimed that the Bayesian network (BN) minimizing CMDL as a discriminative model showed better accuracy than that maximizing ML. Unfortunately, the CLL has no closed-form equation for estimating the optimal parameters. This implies that optimizing the CLL requires a gradient descent algorithm (e.g., extended logistic regression algorithm [4]). Nevertheless, the optimization algorithm involves the reiteration of each structure candidate, which renders the method computationally expensive. To solve this problem, Friedman et al. [2] proposed an augmented naive Bayes classifier (ANB) in which the class variable directly links to all feature variables, and links among feature variables are allowed. ANB ensures that all feature variables can contribute to classification. Later, various types of restricted ANBs were proposed, such as tree-augmented naive Bayes (TAN) [2] and forest-augmented naive Bayes (FAN) [5].

Because maximization of CLL entails heavy computation, various approximation methods have been proposed to maximize it. Carvalho et al. [6] proposed the approximated CLL (aCLL), which is decomposable and computationally efficient. Grossman and Domingos [3] proposed the BNC2P, which is a greedy learning method with at most two parents per variable using the hill-climbing search by maximizing CLL while estimating parameters by maximizing LL. Mihaljević et al. [7] proposed MC-DAGGES, which reduces the space for the greedy search of BN Classifiers (BNCs) using the CLL score. These reports described that the BNC maximizing the approximated CLL performed better than that maximizing the approximated ML. Nevertheless, they did not explain why CLL outperformed ML. For large data, the classification accuracies presented by maximizing ML are expected to be comparable to those presented by maximizing CLL, because ML has asymptotic consistency. Differences between the performances of the two scores in these studies might depend on their respective learning algorithms; they were approximate learning algorithms, not exact ones.

Recent studies have explored efficient algorithms for the exact learning of GBN to maximize ML [8,9,10,11,12,13,14,15,16].

This study compares the classification performances of the BNC with exact learning using ML as a generative model and those with approximate learning using CLL as a discriminative model. The results show that maximizing ML shows better classification accuracy when compared with maximizing CLL for large data. However, the results also show that classification accuracies obtained by exact learning BNC using ML are much worse than those obtained by other methods when the sample size is small and the class variable has numerous parents in the exactly learned networks. When a class variable has numerous parents, estimation of the conditional probability parameters of the class variable become unstable because the number of parent configurations becomes large and the sample size for learning the parameters becomes sparse.

To solve this problem, this study proposes an exact learning ANB which maximizes ML and ensures that the class variable has no parents. In earlier studies, the ANB constraint was used to learn the BNC as a discriminative model. In contrast, we use the ANB constraint to learn the BNC as a generative model. The proposed method asymptotically learns the optimal ANB, which asymptotically represents the true probability distribution with the fewest parameters among all possible ANB structures. Moreover, the proposed ANB is guaranteed to asymptotically estimate the identical conditional probability of the class variable to that of the exactly learned GBN. Furthermore, learning ANBs has lower computational costs than learning GBNs. Although the main theorem assumes that all feature variables are included in the Markov blanket of the class variable, this assumption does not necessarily hold. To address this problem, we propose a feature selection method using Bayes factor for exact learning of the ANB so as to avoid increasing the computational costs. Comparison experiments show that our method outperforms the other methods.

## 2. Background

In this section, we introduce the notation and background material required for our discussion.

### 2.1. Bayesian Network

A BN is a graphical model that represents conditional independence among random variables as a directed acyclic graph (DAG). The BN provides a good approximation of the joint probability distribution because it decomposes the distribution exactly into a product of the conditional probability for each variable.

Let $V=\left\{X_{0},X_{1},\cdots ,X_{n}\right\}$ be a set of discrete variables, where $X_{i},\left(i=0,\cdots ,n\right)$ can take values in the set of states $\left\{1,\cdots ,r_{i}\right\}$. One can say $X_{i}=k$ when $X_{i}$ takes the state *k*. According to the BN structure *G*, the joint probability distribution is represented as
$$\begin{array}{c} P\left(X_{0},X_{1},\cdots ,X_{n}|G\right)=\prod_{i=0}^{n}P\left(X_{i}|{\mathrm{Pa}}_{i}^{G},G\right), \end{array}$$
where ${\mathrm{Pa}}_{i}^{G}$ is the parent variable set of $X_{i}$ in *G*. When the structure *G* is obvious from the context, we use ${\mathrm{Pa}}_{i}$ to denote the parents. Let $\theta_{ijk}$ be a conditional probability parameter of $X_{i}=k$ when the *j*-th instance of the parents of $X_{i}$ is observed (we can say ${\mathrm{Pa}}_{i}=j$). Then, we define $\Theta_{ij}={⋃}_{k=1}^{r_{i}}\left\{\theta_{ijk}\right\},\Theta ={⋃}_{i=0}^{n}{⋃}_{j=1}^{q^{{\mathrm{Pa}}_{i}}}\left\{\Theta_{ij}\right\}$, where $q^{{\mathrm{Pa}}_{i}}=\prod_{v:X_{v}\in {\mathrm{Pa}}_{i}}r_{v}$. A BN is a pair B=(G,Θ).

The BN structure represents conditional independence assertions in the probability distribution by *d-separation*. First, we define *collider*, for which we need to define the d-separation. Letting *path* denote a sequence of adjacent variables, the collider is defined as follows.

**Definition** **1.***Assuming we have a structure*G=(V,E), *a variable*Z∈V*on a path ρ is a collider if and only if there exist two distinct incoming edges into Z from non-adjacent variables*.

We then define d-separation as explained below.

**Definition** **2.***Assuming we have a structure*G=(V,E), X,Y∈V, and Z⊆V∖{X,Y}, *the two variables X and Y are d-separated, given*Z*in G, if and only if every path ρ between X and Y satisfies either of the following two conditions:**Z includes a non-collider on ρ.**There is a collider Z on ρ; Z does not include Z and its descendants.**We denote the d-separation between X and Y given*Z*in the structure G as*$Dsep_{G}\left(X,Y|Z\right)$. *Two variables are d-connected if they are not d-separated*.

If we have X,Y,Z∈V, and *X* and *Y* are not adjacent, then the following three possible types of connections characterize the d-separations: serial connections such as X→Z→Y, divergence connections such as X←Z→Y, and convergence connections such as X→Z←Y. The following theorem of d-separations for these connections holds.

**Theorem** **1**(Koller and Friedman [17]). *First, assume a structure G=(V,E), X,Y,Z∈V. If G has a convergence connection X→Z←Y, then the following two propositions hold:*
$\forall Z\subseteq V\setminus \left\{X,Y,Z\right\},\neg Dsep_{G}\left(X,Y|Z,Z\right),$$\exists Z\subseteq V\setminus \left\{X,Y,Z\right\},Dsep_{G}\left(X,Y|Z\right).$
*If G has a serial connection*
X→Z→Y or divergence connection X←Z→Y, *then negations of the above two propositions hold*.

The two DAGs are *Markov equivalent* when they have the same d-separations.

**Definition** **3.**
*Let $G_{1}=\left(V,E_{1}\right)$ and $G_{2}=\left(V,E_{2}\right)$ be the two DAGs; then $G_{1}$ and $G_{2}$ are called Markov equivalent if the following holds:*

$$\begin{array}{c} \forall X,Y\in V,\forall Z\subseteq V\setminus \left\{X,Y\right\},\;Desp_{G_{1}}\left(X,Y|Z\right)\Leftrightarrow Dsep_{G_{2}}\left(X,Y|Z\right). \end{array}$$



Verma and Pearl [18] described the following theorem to identify Markov equivalence.

**Theorem** **2**(Verma and Pearl [18]). *Two DAGs are Markov equivalent if and only if they have identical links (edges without direction) and identical convergence connections.*

Let $I_{P^{*}}\left(X,Y|Z\right)$ denote that *X* and *Y* are conditionally independent given Z in the true joint probability distribution $P^{*}$. A BN structure *G* is an *independence map (I-map)* if all the d-separations in *G* are entailed by conditional independences in $P^{*}$:

**Definition** **4.**
*Assuming the true joint probability distribution $P^{*}$ of the random variables in a set V and a structure G=(V,E), then G is an I-map if the following proposition holds:*

$$\forall X,Y\in V,\forall Z\subseteq V\setminus \left\{X,Y\right\},Dsep_{G}\left(X,Y|Z\right)\Rightarrow I_{P^{*}}\left(X,Y|Z\right).$$



Probability distributions represented by an I-map converge to $P^{*}$ when the sample size becomes sufficiently large.

We introduce the following notations required for our discussion on learning BNs. Let $D=\{x^{1},\cdots ,x^{d},\cdots ,x^{N}\}$ be a complete dataset consisting of *N* i.i.d. instances, where each instance $x^{d}$ is a data-vector $\left(x_{0}^{d},x_{1}^{d},\cdots ,x_{n}^{d}\right)$. For a variable set Z⊆V, we define $N_{j}^{Z}$ as the number of samples of Z=j in the entire dataset *D*, and we define $N_{ijk}^{Z}$ as the number of samples of $X_{i}=k$ when Z=j in *D*. In addition, we define a joint frequency table JFT(Z) and a conditional frequency table $CFT(X_{i},Z)$, respectively, as a list of $N_{j}^{Z}$ for $j=1,\cdots ,q^{Z}$ and that of $N_{ijk}^{Z}$ for $i=0,\cdots ,n,j=1,\cdots ,q^{Z},$ and $k=1,\cdots ,r_{i}$.

The likelihood of BN *B*, given *D*, is represented as
$$\begin{array}{c} P\left(D|B\right)=\prod_{d=1}^{N}P\left(x_{0}^{d},x_{1}^{d},\cdots ,x_{n}^{d}|B\right)=\prod_{i=0}^{n}\prod_{j=1}^{q^{{\mathrm{Pa}}_{i}}}\prod_{k=1}^{r_{i}}\theta_{ijk}^{N_{ijk}^{{\mathrm{Pa}}_{i}}}, \end{array}$$
where $P(x_{0}^{d},x_{1}^{d},\cdots ,x_{n}^{d}|B)$ represents $P(X_{0}=x_{0}^{d},X_{1}=x_{1}^{d},\cdots ,X_{n}=x_{n}^{d}|B)$. The maximum likelihood estimators of $\theta_{ijk}$ are given as
$$\begin{array}{c} {\hat{\theta }}_{ijk}=\frac{N_{ijk}^{{\mathrm{Pa}}_{i}}}{N_{j}^{{\mathrm{Pa}}_{i}}}. \end{array}$$

The most popular parameter estimator of BNs is the *expected a posteriori* (EAP) of Equation (Equation 1), which is the expectation of $\theta_{ijk}$ with respect to the density $p(\Theta_{ij}|D,G)$ of Equation (Equation 2), assuming Dirichlet prior density $p(\Theta_{ij}|G)$ of Equation (Equation 3).
(1)$$\begin{array}{cc} {\hat{\theta }}_{ijk} & =E\left(\theta_{ijk}|D,G\right)=\int \theta_{ijk}\cdot p\left(\Theta_{ij}|D,G\right)d\Theta_{ij}=\frac{N_{ijk}^{\prime }+N_{ijk}^{{\mathrm{Pa}}_{i}}}{N_{ij}^{\prime }+N_{j}^{{\mathrm{Pa}}_{i}}}. \end{array}$$
(2)$$\begin{array}{c} p\left(\Theta_{ij}|D,G\right)=\frac{\Gamma (\sum_{k=1}^{r_{i}}\left(N_{ijk}^{\prime }+N_{ijk}^{{\mathrm{Pa}}_{i}}\right))}{\prod_{k=1}^{r_{i}}\Gamma \left(N_{ijk}^{\prime }+N_{ijk}^{{\mathrm{Pa}}_{i}}\right)}\prod_{k=1}^{r_{i}}\theta_{ijk}^{N_{ijk}^{\prime }+N_{ijk}^{{\mathrm{Pa}}_{i}}-1}. \end{array}$$
(3)$$\begin{array}{c} p\left(\Theta_{ij}|G\right)=\frac{\Gamma (\sum_{k=1}^{r_{i}}N_{ijk}^{\prime })}{\prod_{k=1}^{r_{i}}\Gamma \left(N_{ijk}^{\prime }\right)}\prod_{k=1}^{r_{i}}\theta_{ijk}^{N_{ijk}^{\prime }-1}. \end{array}$$
In Equations (Equation 1)–(Equation 3), $N_{ijk}^{\prime }$ denotes the hyperparameters of the Dirichlet prior distributions ($N_{ijk}^{\prime }$ is a pseudo-sample corresponding to $N_{ijk}^{{\mathrm{Pa}}_{i}}$), with $N_{ij}^{\prime }=\sum_{k=1}^{r_{i}}N_{ijk}^{\prime }$.

The BN structure must be estimated from observed data because it is generally unknown. To learn the I-map with the fewest parameters, we maximize the score with an *asymptotic consistency* defined as shown below.

**Definition** **5**(Chickering [19]). *Let*
$G_{1}=\left(V,E_{1}\right)$ and $G_{2}=\left(V,E_{2}\right)$
*be the structures. A scoring criterion*
Score
*has an asymptotic consistency if the following two properties hold when the sample size is sufficiently large*.
*If $G_{1}$ is an I-map and $G_{2}$ is not an I-map, then $Score\left(G_{1}\right)>Score\left(G_{2}\right)$.**If $G_{1}$ and $G_{2}$ both are I-maps, and if $G_{1}$ has fewer parameters than $G_{2}$, then $Score\left(G_{1}\right)>Score\left(G_{2}\right)$.*

The ML score P(D∣G) is known to have asymptotic consistency [19].

When we assume the Dirichlet prior density of Equation (Equation 3), ML is represented as
$$\begin{array}{c} P\left(D|G\right)=\prod_{i=0}^{n}\prod_{j=1}^{q^{{\mathrm{Pa}}_{i}}}\frac{\Gamma (N_{ij}^{\prime })}{\Gamma (N_{ij}^{\prime }+N_{j}^{{\mathrm{Pa}}_{i}})}\prod_{k=1}^{r_{i}}\frac{\Gamma (N_{ijk}^{\prime }+N_{ijk}^{{\mathrm{Pa}}_{i}})}{\Gamma (N_{ijk}^{\prime })}. \end{array}$$

In particular, Heckerman et al. [20] presented the following constraint related to the hyperparameters $N_{ijk}^{\prime }$ for ML satisfying the *score-equivalence assumption*, where it takes the same value for the Markov equivalent structures:$$N_{ijk}^{\prime }=N^{\prime }P\left(X_{i}=k,{\mathrm{Pa}}_{i}=j|G^{h}\right),$$
where $N^{\prime }$ is the equivalent sample size (ESS) determined by users, and $G^{h}$ is the hypothetical BN structure that reflects a user’s prior knowledge. This metric was designated as the *Bayesian Dirichlet equivalent* (BDe) score metric. As Buntine [21] described, $N_{ijk}^{\prime }=N^{\prime }/\left(r_{i}q^{{\mathrm{Pa}}_{i}}\right)$ is regarded as a special case of the BDe score. Heckerman et al. [20] called this special case the *Bayesian Dirichlet equivalent uniform* (BDeu), defined as
$$P\left(D|G\right)=\prod_{i=0}^{n}\prod_{j=1}^{q^{{\mathrm{Pa}}_{i}}}\frac{\Gamma (N^{\prime }/q^{{\mathrm{Pa}}_{i}})}{\Gamma (N^{\prime }/q^{{\mathrm{Pa}}_{i}}+N_{j}^{{\mathrm{Pa}}_{i}})}\prod_{k=1}^{r_{i}}\frac{\Gamma (N^{\prime }/\left(r_{i}q^{{\mathrm{Pa}}_{i}}\right)+N_{ijk}^{{\mathrm{Pa}}_{i}})}{\Gamma (N^{\prime }/\left(r_{i}q^{{\mathrm{Pa}}_{i}}\right))}.$$

In addition, the *minimum description length* (MDL) score presented in (Equation 4), which approximates the negative logarithm of ML, is often used for learning BNs.
(4)$$MDL\left(B|D\right)=\frac{\log N}{2}\left|\Theta \right|-\sum_{d=1}^{N}\log P\left(x_{0}^{d},x_{1}^{d},\cdots ,x_{n}^{d}|B\right).$$
The first term of Equation (Equation 4) is the penalty term, which signifies the model complexity. The second term, LL, is the fitting term that reflects the degree of model fitting to the training data.

Both BDeu and MDL are *decomposable*, i.e., the scores can be expressed as a sum of *local scores* depending only on the conditional frequency table for one variable and its parents as follows.
$$\begin{array}{c} Score\left(G\right)=\sum_{i=0}^{n}Score_{i}\left({\mathrm{Pa}}_{i}\right)=\sum_{i=0}^{n}Score\left(CFT\left(X_{i},{\mathrm{Pa}}_{i}\right)\right). \end{array}$$
For example, the local score of log BDeu for $CFT(X_{i},{\mathrm{Pa}}_{i})$ is
(5)$$\begin{array}{c} Score_{i}\left({\mathrm{Pa}}_{i}\right)=\sum_{j=1}^{q^{{\mathrm{Pa}}_{i}}}\left(\log\frac{\Gamma (N^{\prime }/q^{{\mathrm{Pa}}_{i}})}{\Gamma (N^{\prime }/q^{{\mathrm{Pa}}_{i}}+N_{j}^{{\mathrm{Pa}}_{i}})}\sum_{k=1}^{r_{i}}\log\frac{\Gamma (N^{\prime }/\left(r_{i}q^{{\mathrm{Pa}}_{i}}\right)+N_{ijk}^{{\mathrm{Pa}}_{i}})}{\Gamma (N^{\prime }/\left(r_{i}q^{{\mathrm{Pa}}_{i}}\right))}\right). \end{array}$$
The decomposable score enables an extremely efficient search for structures [10,15].

### 2.2. Bayesian Network Classifiers

A BNC can be interpreted as a BN for which $X_{0}$ is the class variable and $X_{1},\cdots ,X_{n}$ are feature variables. Given an instance $x=(x_{1},\cdots ,x_{n})$ for feature variables $X_{1},\ldots ,X_{n}$, the BNC *B* infers class *c* by maximizing the posterior probability of $X_{0}$ as
(6)$$\begin{array}{cc} \hat{c} & =\underset{c\in \{1,\cdots ,r_{0}\}}{\arg\;\max}P\left(c|x_{1},\cdots ,x_{n},B\right)\\ \; & =\underset{c\in \{1,\cdots ,r_{0}\}}{\arg\;\max}\prod_{i=0}^{n}\prod_{j=1}^{q^{{\mathrm{Pa}}_{i}}}\prod_{k=1}^{r_{i}}{\left(\theta_{ijk}\right)}^{1_{ijk}}\\ \; & =\underset{c\in \{1,\cdots ,r_{0}\}}{\arg\;\max}\prod_{j=1}^{q^{{\mathrm{Pa}}_{0}}}\prod_{k=1}^{r_{0}}{\left(\theta_{0jk}\right)}^{1_{0jk}}\times \prod_{i:X_{i}\in C}\prod_{j=1}^{q^{{\mathrm{Pa}}_{i}}}\prod_{k=1}^{r_{i}}{\left(\theta_{ijk}\right)}^{1_{ijk}}, \end{array}$$
where $1_{ijk}=1$ if $X_{i}=k$ and ${\mathrm{Pa}}_{i}=j$ in the case of x, and $1_{ijk}=0$ otherwise. Furthermore, C is a set of children of the class variable $X_{0}$. From Equation (Equation 6), we can infer class *c* given only the values of the parents of $X_{0}$, the children of $X_{0}$, and the parents of the children of $X_{0}$, which comprise the *Markov blanket* of $X_{0}$.

However, Friedman et al. [2] reported that BNC-minimizing MDL cannot optimize classification performance. They proposed the sole use of the following CLL of the class variable given feature variables, instead of the LL for learning BNC structures.
(7)$$\begin{array}{cc} CLL(B & |D)=\sum_{d=1}^{N}\log P\left(x_{0}^{d}|x_{1}^{d},\cdots ,x_{n}^{d},B\right)\\ \; & =\sum_{d=1}^{N}\log P\left(x_{0}^{d},x_{1}^{d},\cdots ,x_{n}^{d}|B\right)-\sum_{d=1}^{N}\log\sum_{c=1}^{r_{0}}P\left(c,x_{1}^{d},\cdots ,x_{n}^{d}|B\right). \end{array}$$
Furthermore, they proposed conditional MDL (CMDL), which is a modified MDL replacing LL with CLL, as shown below.
$$CMDL\left(B|D\right)=\frac{\log N}{2}\left|\Theta \right|-CLL\left(B|D\right).$$

Consequently, they claimed that the BN minimizing CMDL as a discriminative model showed better accuracy than that maximizing ML as a generative model.

Unfortunately, the CLL is not decomposable, because we cannot describe the second term of Equation (Equation 7) as a sum of the log parameters in Θ. This finding implies that no closed-form equation exists for the maximum CLL estimator for Θ. Therefore, learning the network structure that minimizes the CMDL requires a search method such as gradient descent over the space of parameters for each structure candidate. Therefore, exact learning network structures by minimizing CMDL is computationally infeasible.

As a simple means of resolving that difficulty, Friedman et al. [2] proposed an ANB that ensures an edge from the class variable to each feature variable and allows edges among feature variables. Furthermore, they proposed TAN in which the class variable has no parent, and each feature variable has a class variable and at most one other feature variable as a parent variable.

Various approximate methods to maximize CLL have been proposed. Carvalho et al. [6] proposed an aCLL score, which is decomposable and computationally efficient. Let $G_{ANB}$ be an ANB structure. In addition, let $N_{ijck}$ be the number of samples of $X_{i}=k$ when $X_{0}=c$ and ${\mathrm{Pa}}_{i}\setminus \left\{X_{0}\right\}=j$$\left(i=1,\cdots ,n;j=1,\cdots ,q^{{\mathrm{Pa}}_{i}\setminus \left\{X_{0}\right\}};c=1,\cdots ,r_{0};k=1,\cdots ,r_{i}\right)$. In addition, let $N^{\prime \prime }>0$ represent the number of pseudocounts. Under several assumptions, the aCLL can be represented as   
$$\begin{array}{c} aCLL\left(G_{ANB}|D\right)\propto \sum_{i=1}^{n}\sum_{j=1}^{q^{{\mathrm{Pa}}_{i}\setminus \left\{X_{0}\right\}}}\sum_{k=1}^{r_{i}}\sum_{c=1}^{r_{0}}\left(N_{ijck}+\beta \sum_{c^{\prime }=1}^{r_{0}}N_{ijc^{\prime }k}\right)\log\frac{N_{ij+ck}}{N_{ij+c}}, \end{array}$$
where
$$N_{ij+ck}=\left\{\begin{array}{c} N_{ijck}+\beta \sum_{c^{\prime }=1}^{r_{0}}N_{ijc^{\prime }k}\;\;\;\mathrm{if}\;N_{ijck}+\beta \sum_{c^{\prime }=1}^{r_{0}}N_{ijc^{\prime }k}\geq N^{\prime \prime }\\ N^{\prime \prime }\;\;\;\;\;\;\;\;\;\;\;\;\;\;\;\;\;\;\;\;\mathrm{otherwise}, \end{array}\right.$$
$$N_{ij+c}=\sum_{k=1}^{r_{i}}N_{ij+ck}.$$
The value of β is found by using the Monte Carlo method to approximate the CLL. When the value of β is optimal, the aCLL is a minimum-variance unbiased approximation of the CLL.

Moreover, Grossman and Domingos [3] proposed a learning structure method using a greedy hill-climbing algorithm [20] by maximizing the CLL while estimating the parameters by maximizing the LL. Recently, Mihaljević et al. [7] identified the smallest subspace of DAGs that covered all possible class-posterior distributions when the data were complete.

All the DAGs in this space, which they call *minimal class-focused* DAGs (MC-DAGs), are such that every edge is directed toward a child of the class variable. In addition, they proposed a greedy search algorithm in the space of Markov equivalent classes of MC-DAGs using the CLL score. These reports described that the BNC maximizing the approximated CLL provides better performance than that maximizing the approximated ML. However, they did not explain why CLL outperformed ML. For large data, the classification accuracies obtained by maximizing ML are expected to be comparable to those obtained by maximizing CLL because ML has asymptotic consistency. Differences between the performances of the two scores in these earlier studies might depend on their learning algorithms to maximize ML; they were approximate learning algorithms, not exact ones.

## 3. Classification Accuracies of Exact Learning GBN

This section presents experiments comparing the classification accuracies of the exactly learned GBN by maximizing the BDeu as a generative model with those of the approximately learned BNC by maximizing the CLL as a discriminative model. Although determining the hyperparameter $N^{\prime }$ of BDeu is difficult [16,22,23,24], we use $N^{\prime }=1.0$, which allows the data to reflect the estimated parameters to the greatest degree possible [25,26].

The experiment compared the respective classification accuracies of seven methods in Table 1. All the methods were implemented in Java. The source code is available at http://www.ai.lab.uec.ac.jp/software/ (accessed on 29 November 2021). Throughout this paper, our experiments were conducted on a computational environment, as shown in Table 2. This experiment used 43 classification benchmark datasets from the *UCI repository* [27]. Continuous variables were discretized into two bins using the median value as the cutoff, as in [28]. In addition, data with missing values were removed from the datasets. We used EAP estimators as conditional probability parameters of the respective classifiers. The hyperparameters $N_{ijk}^{\prime }$ of EAP were found to be $1/(r_{i}q^{{\mathrm{Pa}}_{i}})$. Throughout our experiments, we defined “small datasets” as the datasets with less than 200 samples, and we defined “large datasets” as the datasets with 10,000 or more samples.

Table 3 presents the classification accuracies of the respective classifiers. However, we will discuss the results of *ANB-BDeu* and *fsANB-BDeu* in a later section. The values shown in bold in Table 3 represent the best classification accuracies for each dataset. Here, the classification accuracies represent the average percentage of correct classifications from a ten-fold cross-validation. Moreover, to investigate the relation between the classification accuracies and *GBN-BDeu*, Table 4 presents the details of the achieved structures using *GBN-BDeu*. “Parents” in Table 4 represents the average number of the class variable’s parents in the structures learned by *GBN-BDeu*. “Children” denotes the average number of the class variable’s children in the structures learned by *GBN-BDeu*. “Sparse data” denotes the average number of patterns of $X_{0}$’s parents value *j* with null data, $N_{j}^{{\mathrm{Pa}}_{0}}=0$$\left(j=1,\cdots ,q^{{\mathrm{Pa}}_{0}}\right)$ in the structures learned by *GBN-BDeu*.

From Table 3, *GBN-BDeu* shows the best classification accuracies among the methods for large data, such as dataset Nos. 22, 29, and 33. Because BDeu has asymptotic consistency, the joint probability distribution represented by *GBN-BDeu* approaches the true distribution as the sample size increases. However, it is worth noting that *GBN-BDeu* provides much worse accuracy than the other methods in datasets No. 3 and No. 9. In these datasets, the learned class variables by *GBN-BDeu* have no children. Numerous parents are shown in “Parents” and “Children” in Table 4. When a class variable has numerous parents, the estimation of the conditional probability parameters of the class variable becomes unstable, because the class variable’s parent configurations become numerous. Then, the sample size for learning the parameters becomes small, as presented in “Sparse data” in Table 4. Therefore, numerous parents of the class variable might be unable to reflect the feature data for classification when the sample is insufficiently large.

## 4. Exact Learning ANB Classifier

The preceding section suggested that exact learning of GBN by maximizing BDeu to have no parents of the class variable might improve the accuracy of *GBN-BDeu*. In this section, we propose an exact learning ANB, which maximizes BDeu and ensures that the class variable has no parents. In earlier reports, the ANB constraint was used to learn the BNC as a discriminative model. In contrast, we use the ANB constraint to learn the BNC as a generative model. The space of all possible ANB structures includes at least one I-map, because it includes a complete graph, which is an I-map. From the asymptotic consistency of BDeu (Definition 5), the proposed method is guaranteed to achieve the I-map with the fewest parameters among all possible ANB structures when the sample size becomes sufficiently large. Our empirical analysis in Section 3 suggests that the proposed method can improve the classification accuracy for small data. We employed the dynamic programming (DP) algorithm learning GBN [10] for the exact learning of ANB. The DP algorithm for exact learning ANB was almost twice as fast as that for the exact learning of GBN. We prove that the proposed ANB asymptotically estimates the identical conditional probability of the class variable to that of the exactly learned GBN.

### 4.1. Learning Procedure

The proposed method is intended to seek the optimal structure that maximizes the BDeu score among all possible ANB structures. The local score of the class variable in ANB structures is constant because the class variable has no parents in the ANB structure. Therefore, we can ascertain the optimal ANB structure by maximizing $Score_{ANB}\left(G\right)=Score\left(G\right)-Score_{0}\left(\phi \right)$.

Before we describe the procedure of our method, we introduce the following notations. Let $G^{*}\left(Z\right)$ denote the optimal ANB structure composed of a variable set $Z,(X_{0}\in Z)$. When a variable has no child in a structure, we say it is a *sink* in the structure. We use $X_{s}^{*}\left(Z\right)$ to denote a sink in $G^{*}\left(Z\right)$. Additionally, letting Π(Z) denote a set of all the Z’s subsets including $X_{0}$, we define the *best parents* of $X_{i}$ in a candidate set Π(Z), as the parent set that maximizes the local score in Π(Z):$$g_{i}^{*}\left(\Pi \left(Z\right)\right)=\underset{W\in \Pi (Z)}{\arg\;\max}\;Score_{i}\left(W\right).$$

Our algorithm has four logical steps. The following process improves the DP algorithm proposed by [10] to learn the optimal ANB structure.
(1)For all possible pairs of a variable $X_{i}\in V\setminus \left\{X_{0}\right\}$ and a variable set $Z\subseteq V\setminus \left\{X_{i}\right\},\left(X_{0}\in Z\right)$, calculate the local score $Score_{i}\left(Z\right)$ (Equation (Equation 5)).(2)For all possible pairs of a variable $X_{i}\in V\setminus \left\{X_{0}\right\}$ and a variable set $Z\subseteq V\setminus \left\{X_{i}\right\},\left(X_{0}\in Z\right)$, calculate the best parents $g^{*}\left(\Pi \left(Z\right)\right)$.(3)For $\forall Z\subseteq V,(X_{0}\in Z)$, calculate the sink $X_{s}^{*}\left(Z\right)$.(4)Calculate $G^{*}\left(V\right)$ using Steps 3 and 4.
Steps 3 and 4 of the algorithm are based on the observation that the best network $G^{*}\left(Z\right)$ necessarily has a sink $X_{s}^{*}\left(Z\right)$ with incoming edges from its best parents $g_{s}^{*}\left(\Pi \left(Z\setminus \left\{X_{s}^{*}\left(Z\right)\right\}\right)\right)$. The remaining variables and edges in $G^{*}\left(Z\right)$ necessarily construct the best network $G^{*}\left(Z\setminus \left\{X_{s}^{*}\left(Z\right)\right\}\right)$. More formally,
(8)$$\begin{array}{c} X_{s}^{*}\left(Z\right)=\underset{X_{i}\in Z\setminus \left\{X_{0}\right\}}{\arg\;\max}\left\{Score_{i}\left(g_{i}^{*}\left(\Pi \left(Z\setminus \left\{X_{i}\right\}\right)\right)\right)+Score_{ANB}\left(G^{*}\left(Z\setminus \left\{X_{i}\right\}\right)\right)\right\}. \end{array}$$

From Equation (Equation 8), we can decompose $G^{*}\left(Z\right)$ into $G^{*}\left(Z\setminus \left\{X_{s}^{*}\left(Z\right)\right\}\right)$ and $X_{s}^{*}\left(Z\right)$ with incoming edges from $g_{s}^{*}(\Pi \left(Z\setminus \left\{X_{s}^{*}\left(Z\right)\right\}\right)$. Moreover, this decomposition can be performed recursively. At the end of the recursive decomposition, we obtain *n* pairs of the sink and its best network, denoted by $\left(X_{s_{1}},g_{s_{1}}^{*}\right),\cdots ,\left(X_{s_{i}},g_{s_{i}}^{*}\right),\cdots ,\left(X_{s_{n}},g_{s_{n}}^{*}\right)$. Finally, we obtain $G^{*}\left(V\right)$ for which $X_{s_{i}}$’s parent set is $g_{s_{i}}^{*}$.

The number of iterations to calculate all the local scores, best parents, and best sinks for our algorithm are $\left(n-1\right)2^{n-2}$, $\left(n-1\right)2^{n-2}$, and $2^{n-1}$, respectively, and those for GBN are $n2^{n-1}$, $n2^{n-1}$, and $2^{n}$, respectively. Therefore, the DP algorithm for ANB is almost twice as fast as that for GBN. The details of the proposed algorithm are shown in the Appendix A.

### 4.2. Asymptotic Properties of the Proposed Method

Under some assumptions, the proposed ANB is proven to asymptotically estimate the identical conditional probability of the class variable, given the feature variables of the exactly learned GBN. When the sample size becomes sufficiently large, the structure learned by the proposed method and the exactly learned GBN are *classification-equivalent* defined as follows:

**Definition** **6**(Acid et al. [29]). *Let G be a set of all the BN structures. Furthermore, let D be any finite dataset. For $\forall G_{1},G_{2}\in G$, we say that $G_{1}$ and $G_{2}$ are classification-equivalent if $P\left(X_{0}|x,G_{1},D\right)=P\left(X_{0}|x,G_{2},D\right)$ for any feature variable’s value x.*

To derive the main theorem, we introduce five lemmas as below.

**Lemma** **1**(Mihaljević et al. [7]). *Let*
G=(V,E)
*be a structure. Then, G is classification-equivalent to*
$G^{\prime }$, *which is a modified G by the following operations*:
(1)*For $\forall X,Y\in {\mathrm{Pa}}_{0}^{G}$, add an edge between X and Y in G.*(2)*For $\forall X\in {\mathrm{Pa}}_{0}^{G}$, reverse an edge from X to $X_{0}$ in G.*

Next, we use the following lemma from Chickering [19] to derive the main theorem:

**Lemma** **2**(Chickering [19]). *Let $G^{Imap}$ be a set of all I-maps. When the sample size becomes sufficiently large, then the following proposition holds.*
$$\begin{array}{c} \forall G_{1},G_{2}\in G^{Imap},(\left(\forall X,Y\in V,\forall Z\subseteq V\setminus \left\{X,Y\right\},Dsep_{G_{1}}\left(X,Y|Z\right)\Rightarrow Dsep_{G_{2}}\left(X,Y|Z\right)\right)\\ \Rightarrow Score\left(G_{1}\right)\leq Score\left(G_{2}\right)). \end{array}$$

Moreover, we provide Lemma 3 under the following assumption.

**Assumption** **1.**
*Let the true joint probability distribution of random variables in a set V be $P^{*}$. Under Assumption A1, a true structure $G^{*}=\left(V,E^{*}\right)$ exists that satisfies the following property:*

$$\forall X,Y\in V,\forall Z\subseteq V\setminus \left\{X,Y\right\},Dsep_{G^{*}}\left(X,Y|Z\right)\Leftrightarrow I_{P^{*}}\left(X,Y|Z\right).$$



**Lemma** **3.**
*Let $G_{ANB}^{Imap}$ be a set of all the ANB structures that are I-maps. For $\forall G_{ANB}^{Imap}\in G_{ANB}^{Imap},\forall X,Y\in V$, if $G^{*}$ has a convergence connection $X\to X_{0}\leftarrow Y$, then $G_{ANB}^{Imap}$ has an edge between X and Y.*


**Proof.** We prove Lemma 3 by contradiction. Assuming that $G_{ANB}^{Imap}$ has no edge between *X* and *Y*, because $G_{ANB}^{Imap}$ has a divergence connection $X\leftarrow X_{0}\to Y$, we obtain
(9)$$\begin{array}{c} \exists Z\subseteq V\setminus \left\{X,Y,X_{0}\right\},Dsep_{G_{ANB}^{Imap}}\left(X,Y|X_{0},Z\right). \end{array}$$
Because $G^{*}$ has a convergence connection $X\to X_{0}\leftarrow Y$, the following proposition holds from Theorem 1:
(10)$$\begin{array}{c} \forall Z\subseteq V\setminus \left\{X,Y,X_{0}\right\},\neg Dsep_{G_{ANB}^{Imap}}\left(X,Y|X_{0},Z\right). \end{array}$$ This result contradicts (Equation 9). Consequently, $G_{ANB}^{Imap}$ has an edge between *X* and *Y*.    □

Furthermore, under Assumption A1 and the following assumptions, we derive Lemma 4.

**Assumption** **2.***All feature variables are included in the Markov blanket M of the class variable in the true structure*$G^{*}$.

**Assumption** **3.**
*For ∀X∈M, X and $X_{0}$ are adjacent to $G^{*}$.*


**Lemma** **4.**
*Let $G_{1}^{*}$ be the modified $G^{*}$ by operation 1 in Lemma 1. In addition, let $G_{12}^{*}$ be the structure that is modified to $G_{1}^{*}$ by operation 2 in Lemma 1. Under Assumptions 1–3, $G_{1}^{*}$ is Markov equivalent to $G_{12}^{*}$.*


**Proof.** From Theorem 2, we prove Lemma 4 by showing the following two propositions: (I) $G_{1}^{*}$ and $G_{12}^{*}$ have the same links (edges without direction), and (II) they have the same set of convergence connections. Proposition (I) can be proved immediately because the difference between $G_{1}^{*}$ and $G_{12}^{*}$ is only the direction of the edges between $X_{0}$ and the variables in ${\mathrm{Pa}}_{0}^{G^{*}}$. For the same reason, $G_{1}^{*}$ and $G_{12}^{*}$ have the same set of convergence connections as colliders in $V\setminus ({\mathrm{Pa}}_{0}^{G^{*}}\cup \left\{X_{0}\right\})$. Moreover, there are no convergence connections with colliders in ${\mathrm{Pa}}_{0}^{G^{*}}\cup \left\{X_{0}\right\}$ in both $G_{1}^{*}$ and $G_{12}^{*}$ because all the variables in ${\mathrm{Pa}}_{0}^{G^{*}}\cup \left\{X_{0}\right\}$ are adjacent in the two structures. Consequently, they have the same set of convergence connections, so Proposition (II) holds. This completes the proof.    □

Finally, under Assumptions 1–3, we derive the following lemma.

**Lemma** **5.**
*Under Assumptions 1–3, $G_{12}^{*}$ is an I-map.*


**Proof.** The DAG $G_{1}^{*}$ results from adding the edges between the variables in ${\mathrm{Pa}}_{0}^{G^{*}}$ to $G^{*}$. Because adding edges does not create a new d-separation, $G_{1}^{*}$ remains an I-map. Lemma 5 holds because $G_{1}^{*}$ is a Markov equivalent to $G_{12}^{*}$ from Lemma 4.    □

Under Assumptions 1–3, we prove the following main theorem using Lemmas 1–5.

**Theorem** **3.***Under Assumptions 1–3, when the sample becomes sufficiently large, the proposal (learning ANB using BDeu) achieves the classification-equivalent structure to*$G^{*}$.

**Proof.** Because $G_{12}^{*}$ is classification-equivalent to $G^{*}$ from Lemma 1, we prove Theorem 3 by showing that the proposed method asymptotically learns a Markov-equivalent structure to $G_{12}^{*}$. We prove Theorem 3 by showing that $G_{12}^{*}$ asymptotically has the maximum BDeu score among all the ANB structures:
(11)$$\begin{array}{c} \forall G_{ANB}\in G_{ANB},\;Score\left(G_{ANB}\right)\leq Score\left(G_{12}^{*}\right). \end{array}$$ From Definition 5, the BDeu scores of the I-maps are higher than those of any non-I-maps when the sample size becomes sufficiently large. Therefore, it is sufficient to show that the following proposition holds asymptotically to prove that Proposition (Equation 11) asymptotically holds.
(12)$$\begin{array}{c} \forall G_{ANB}^{Imap}\in G_{ANB}^{Imap},\;Score\left(G_{ANB}^{Imap}\right)\leq Score\left(G_{12}^{*}\right). \end{array}$$ From Lemma 5, $G_{12}^{*}$ is an I-map. Therefore, from Lemma 2, a sufficient condition for satisfying (Equation 12) is as follows:
(13)$$\begin{array}{cc} \forall G_{ANB}^{Imap}\in G_{ANB}^{Imap},\forall X,Y\in M\cup \left\{X_{0}\right\},\\ \forall Z\subseteq M\cup \left\{X_{0}\right\}\setminus \left\{X,Y\right\},\; & Dsep_{G_{ANB}^{Imap}}\left(X,Y|Z\right)\Rightarrow Dsep_{G_{12}^{*}}\left(X,Y|Z\right). \end{array}$$We prove (Equation 13) by dividing it into two cases: $X\in {\mathrm{Pa}}_{0}^{G^{*}}\wedge Y\in {\mathrm{Pa}}_{0}^{G^{*}}$ and $X\notin {\mathrm{Pa}}_{0}^{G^{*}}\vee Y\notin {\mathrm{Pa}}_{0}^{G^{*}}$.
**Case I:**$X\in {\mathrm{Pa}}_{0}^{G^{*}}\wedge Y\in {\mathrm{Pa}}_{0}^{G^{*}}$   From Lemma 3, all variables in ${\mathrm{Pa}}_{0}^{G^{*}}$ are adjacent to $G_{ANB}^{Imap}$. Therefore, we obtain
(14)$$\begin{array}{c} \forall Z\subseteq M\cup \left\{X_{0}\right\}\setminus \left\{X,Y\right\},\neg Dsep_{G_{ANB}^{Imap}}\left(X,Y|Z\right)\wedge \neg Dsep_{G_{12}^{*}}\left(X,Y|Z\right). \end{array}$$   For two Boolean propositions *p* and *q*, the following holds.
(15)$$\begin{array}{c} \left(\neg p\wedge \neg q)\Rightarrow (p\Leftrightarrow q\right) \end{array}$$   From (Equation 14) and (Equation 15), we obtain
$$\begin{array}{c} \forall Z\subseteq M\cup \left\{X_{0}\right\}\setminus \left\{X,Y\right\},\;Dsep_{G_{ANB}^{Imap}}\left(X,Y|Z\right)\Leftrightarrow Dsep_{G_{12}^{*}}\left(X,Y|Z\right). \end{array}$$**Case II**: $X\notin {\mathrm{Pa}}_{0}^{G^{*}}\vee Y\notin {\mathrm{Pa}}_{0}^{G^{*}}$   From Definition 4 and Assumption A1, we obtain
$$\begin{array}{c} \forall Z\subseteq M\cup \left\{X_{0}\right\}\setminus \left\{X,Y\right\},Dsep_{G_{ANB}^{Imap}}\left(X,Y|Z\right)\Rightarrow Dsep_{G^{*}}\left(X,Y|Z\right). \end{array}$$   Thus, we can prove (Equation 13) by showing that the following proposition holds:
(16)$$\begin{array}{c} \forall Z\subseteq M\cup \left\{X_{0}\right\}\setminus \left\{X,Y\right\},Dsep_{G^{*}}\left(X,Y|Z\right)\Leftrightarrow Dsep_{G_{12}^{*}}\left(X,Y|Z\right). \end{array}$$For the remainder of the proof, we prove the sufficient condition (Equation 16) to satisfy (Equation 13) by dividing it into two cases: $X_{0}\in Z$ and $X_{0}\notin Z$.   **Case i**: $X_{0}\in Z$
All pairs of non-adjacent variables in ${\mathrm{Pa}}_{0}^{G^{*}}$ in $G^{*}$ comprise a convergence connection with collider $X_{0}$. From Theorem 1, these pairs are necessarily d-connected, given $X_{0}$ in $G^{*}$. Therefore, all the variables in ${\mathrm{Pa}}_{0}^{G^{*}}$ are d-connected, given $X_{0}$ in $G^{*}$. This means that $G^{*}$ and $G_{1}^{*}$ represent identical d-separations given $X_{0}$. Because $G_{1}^{*}$ is Markov equivalent to $G_{12}^{*}$ from Lemma 4, $G^{*}$ and $G_{12}^{*}$ represent identical d-separations given $X_{0}$; i.e., Proposition (Equation 16) holds.   **Case ii**: $X_{0}\notin Z$      We divide (Equation 16) into two cases: $X=X_{0}\vee Y=X_{0}$ and $X\neq X_{0}\wedge Y\neq X_{0}$      **Case 1**: $X=X_{0}\vee Y=X_{0}$
Because all the variables in the $X_{0}$’s Markov blanket *M* are adjacent to $X_{0}$ in both $G_{12}^{*}$ and $G^{*}$ from Assumption A2, we obtain $\neg Dsep_{G_{12}^{*}}\left(X,Y|Z\right)\wedge \neg Dsep_{G^{*}}\left(X,Y|Z\right)$. From (Equation 15), proposition (Equation 16) holds.      **Case 2**: $X\neq X_{0}\wedge Y\neq X_{0}$
If both $G_{12}^{*}$ and $G^{*}$ have no edge between *X* and *Y*, they have a serial or divergence connection: $X\to X_{0}\to Y$ or $X\leftarrow X_{0}\to Y$. Because the serial and divergence connections represent d-connections between *X* and *Y* in this case from Theorem 1, we obtain $\neg Dsep_{G_{12}^{*}}\left(X,Y|Z\right)\wedge \neg Dsep_{G^{*}}\left(X,Y|Z\right)$. From (Equation 15), proposition (Equation 16) holds.   Thus, we complete the proof of (Equation 13) in **Case II**.
Consequently, proposition (Equation 13) is true, which completes the proof of Theorem 3.    □

We proved that the proposed ANB asymptotically estimates the identical conditional probability of the class variable to that of the exactly learned GBN.

### 4.3. Numerical Examples

This subsection presents the numerical experiments conducted to demonstrate the asymptotic properties of the proposed method. To demonstrate that the proposed method asymptotically achieves the I-map with the fewest parameters among all the possible ANB structures, we evaluated the structural Hamming distance (SHD) [30], which measures the distance between the structure learned by the proposed method and the I-map with the fewest parameters among all the possible ANB structures. To demonstrate Theorem 3, we evaluated the Kullback–Leibler divergence (KLD) between the learned class variable posterior using the proposed method and that by the true structure. This experiment used two benchmark datasets from *bnlearn* [31]: CANCER and ASIA, as depicted in Figure 1 and Figure 2. We used the variables “Cancer” and “either” as the class variables in CANCER and ASIA, respectively. In that case, CANCER satisfied Assumptions 2 and 3, but ASIA did not.

From the two networks, we randomly generated sample data for each sample size *N* = 100, 500, 1000, 5000, 10,000, 50,000, and 100,000. Based on the generated data, we learned the BNC structures using the proposed method and then evaluated the SHDs and KLDs. Table 5 presents the results. The results show that the SHD converged to 0 when the sample size increased in both CANCER and ASIA. Thus, the proposed method asymptotically learned the I-map with the fewest parameters among all possible ANB structures. Furthermore, in CANCER, the KLD between the learned class variable posterior by the proposed method and that by the true structure became 0 when N≥ 1000. The results demonstrate that the proposed method learns the classification-equivalent structure of the true one when the sample size becomes sufficiently large, as described in Theorem 3. In ASIA however, the KLD between the learned class variable posterior by the proposed method and that by the true structure did not reach 0 even when the sample size became large because ASIA did not satisfy Assumptions 2 and 3.

## 5. Learning Markov Blanket

Theorem 3 assumes all feature variables are included in the Markov blanket of the class variable. However, this assumption does not necessarily hold. To solve this problem, we must learn the Markov blanket of the class variable before learning the ANB. Under Assumption 3, the Markov blanket of the class variable is equivalent to the parent-child (PC) set of the class variable. It is known that the exact learning of a PC set of variables is computationally infeasible when the number of variables increases. To reduce the computational cost of learning a PC set, ref. [32] proposed a score-based local learning algorithm (SLL), which has two learning steps. In step 1, the algorithm sequentially learns the PC set by repeatedly using the exact learning structure algorithm on a set of variables containing the class variable, the current PC set, and one new query variable. In step 2, SLL enforces the symmetry constraint: if $X_{i}$ is a child of $X_{j}$, then $X_{j}$ is a parent of $X_{i}$. This allows us to try removing extra variables from PC set, proving that the SLL algorithm always finds the correct PC of the class variable when the sample size is sufficiently large. Moreover, ref. [33] proposed the $S^{2}$ TMB algorithm, which improved the efficiency over the SLL by removing the symmetric constraints in PC search steps. However, $S^{2}$ TMB is computationally infeasible when the size of the PC set surpasses 30.

As an alternative approach for learning large PC sets, previous studies proposed constraint-based PC search algorithms, such as MMPC [30], HITON-PC [34], and PCMB [35]. These methods produce an undirected graph structure using statistical hypothesis tests or information theory tests. As statistical hypothesis tests, the $G^{2}$ and $\chi^{2}$ tests were used for these constraint-based methods. In these tests, the independence of the two variables was set as a null hypothesis. A *p*-value signifies the probability that the null hypothesis is correct at a user-determined significance level. If the *p*-value exceeds the significance level, the null hypothesis is accepted, and the edge is removed. However, [36] reported that statistical hypothesis tests have a significant shortcoming: the *p*-value sometimes becomes much smaller than the significance level as the sample size increases. Therefore, statistical hypothesis tests suffer from Type I errors (detecting dependence for an independent conditional relation in the true DAG). Conditional mutual information (CMI) is often used as a CI test [37]. The CMI strongly depends on a hand-tuned threshold value. Therefore, it is not guaranteed to estimate the true CI structure. Consequently, CI tests have no asymptotic consistency.

For a CI test with asymptotic consistency, [38] proposed a Bayes factor with BDeu (the “BF method”, below), where the Bayes factor is the ratio of marginal likelihoods between two hypotheses [39]. For two variables X,Y∈V and a set of conditional variables Z⊆V∖{X,Y}, the BF method BF(X,Y∣Z) is defined as
$$BF\left(X,Y|Z\right)=\frac{\exp(Score(CFT(X,Z)))}{\exp(Score(CFT(X,Z\cup \{Y\})))},$$
where Score(CFT(X,Z)) and Score(CFT(X,Z∪{Y})) can be obtained using Equation (Equation 5). The BF method detects $I_{P^{*}}\left(X,Y|Z\right)$ if BF(X,Y∣Z) is larger than the threshold δ, and detects the $\neg I_{P^{*}}\left(X,Y|Z\right)$ otherwise. Natori et al. [40] and Natori et al. [41] applied the BF method to a constraint-based approach, and showed that their method was more accurate than the other methods with traditional CI tests.

We propose the constraint-based PC search algorithm using a BF method. The proposed PC search algorithm finds the PC set of the class variable using a BF method between the class variable and all feature variables because the Bayes factor has an asymptotic consistency for the CI tests [41]. It is known that missing crucial variables degrades the accuracy [2]. Therefore, we redundantly learn the PC set of the class variable to reduce extra variables with no missing variables as follows.

The proposed PC search algorithm only conducts the CI tests at the zero order (given no conditional variables), which is more reliable than those at the higher order.We use a positive value as Bayes factor’s threshold δ.

Furthermore, we compare the accuracy of the proposed PC search method with those of the MMPC, HITON-PC, PCMB, and $S^{2}$ TMB. Learning Bayesian networks is known to be highly sensitive to the chosen ESS-value [22,25,26]. Therefore, we determine the ESS $N^{\prime }\in \left\{1.0,2.0,5.0\right\}$ and the threshold δ∈{3,20,150} in the Bayes factor using two-fold cross validation to obtain the highest classification accuracy. The three ESS-values of $N^{\prime }$ are determined according to Ueno [25], Ueno [26]. The three values of δ are determined according to Heckerman et al. [20]. All the compared methods were implemented in Java (Source code is available at http://www.ai.lab.uec.ac.jp/software/, accessed on 29 November 2021). This experiment used six benchmark datasets from *bnlearn*: ASIA, SACHS, CHILD, WATER, ALARM, and BARLEY. From each benchmark network, we randomly generated sample data *N* = 10,000. Based on the generated data, we learned all the variables’ PC sets using each method. Table 6 shows the average runtimes of each method. We calculated missing variables, representing the number of removed variables existing in the true PC set, and extra variables, which indicated the number of remaining variables that do not exist in the true PC set. Table 6 also shows the average missing and extra variables from the learned PC sets of all the variables. We compared the classification accuracies of the exact learning ANB with BDeu score (designated as *ANB-BDeu*), using each PC search method as a feature selection method. Table 7 shows the average accuracies of each method from the 43 UCI repository datasets listed in Table 3.

From Table 6, the results show that the runtimes of the proposed method were shorter than those of the other methods. Moreover, the results show that the missing variables of the proposed method were smaller than those of the other methods. On the other hand, Table 6 also shows that the extra variables of the proposal were greater than those of the other methods in all datasets. From Table 7, the results show that the *ANB-BDeu* using the proposed method provided a much higher average accuracy than the other methods. This is because missing variables degrade classification accuracy more significantly than extra variables (Friedman et al., 1997).

## 6. Experiments

This section presents numerical experiments conducted to evaluate the effectiveness of the exact learning ANB. First, we compared the classification accuracies of *ANB-BDeu* with those of the other methods in Section 3. We used the same experimental setup and evaluation method described in Section 3. The classification accuracies of *ANB-BDeu* are presented in Table 3. To confirm the significant differences of *ANB-BDeu* from the other methods, we applied Hommel’s tests [42], which are used as a standard in machine learning studies [43]. The *p*-values are presented at the bottom of Table 3. In addition, “MB size” in Table 4 denotes the average number of the class variable’s Markov blanket size in the structures learned by *GBN-BDeu*.

The results show that *ANB-BDeu* outperformed *Naive Bayes*, *GBN-CMDL*, *BNC2P*, *TAN-aCLL*, *gGBN-BDeu*, and *MC-DAGGES* at the p<0.1 significance level. Moreover, the results show that *ANB-BDeu* improved the accuracy of *GBN-BDeu* when the class variable had numerous parents such as the No. 3, No. 9, and No. 31 datasets, as shown in Table 4. Furthermore, *ANB-BDeu* provided higher accuracies than *GBN-BDeu*, even for large data such as datasets 13, 22, 29, and 33, although the difference between *ANB-BDeu* and *GBN-BDeu* was not statistically significant. These actual datasets did not necessarily satisfy Assumptions 1–3 in Theorem 3. These results imply that the accuracies of *ANB-BDeu* without satisfying Assumptions 1–3 might be comparable to those of *GBN-BDeu* for large data. It is worth noting that the accuracies of *ANB-BDeu* were much worse than those provided by *GBN-BDeu* for datasets No. 5 and No. 12. “MB size” in these datasets were much smaller than the number of all feature variables, as shown in Table 4. The results show that feature selection by the Markov blanket is expected to improve the classification accuracies of the exact learning ANB, as described in Section 5.

We compared the classification accuracies of *ANB-BDeu* using the PC search method proposed in Section 5 (referred to as “*fsANB-BDe*”) with the other methods in Table 3. Table 3 shows the classification accuracies of *fsANB-BDe* and the *p*-values of Hommel’s tests for differences in *fsANB-BDeu* from the other methods. The results show that *fsANB-BDeu* outperformed all the compared methods at the p<0.05 significance level.

“Max parents” in Table 4 presents the average maximum number of parents learned by *fsANB-BDeu*. The value of “Max parents” represents the complexity of the structure learned by *fsANB-BDeu*. The results show that the accuracies of *Naive Bayes* were better than those of *fsANB-BDeu* when the sample size was small, such as the No. 36 and No. 38 datasets. In these datasets, the values of “Max parents” are large. The estimation of the variable parameters tends to become unstable when a variable has numerous parents, as described in Section 3. *Naive Bayes* can avoid this phenomenon because the maximum number of parents in *Naive Bayes* is one. However, *Naive Bayes* cannot learn relationships between the feature variables. Therefore, for large samples such as the No. 8 and No. 29 datasets, *Naive Bayes* showed much worse accuracy than those provided by other methods.

Similar to *Naive Bayes*, *BNC2P* and *TAN-aCLL* show better accuracies than *fsANB-BDeu* for small samples such as the No. 38 dataset because the upper bound of the maximum number of parents was two in the two methods. However, the small upper bound of the maximum number of parents tends to lead to a poor representational power of the structure [44]. As a result, the accuracies of both methods tend to be worse than those of the *fsANB-BDeu* of which the value of “Max parents” is greater than two, such as the No. 29 dataset.

For large samples, such as datasets Nos 29 and 33, *GBN-CMDL*, *gGBN-BDeu*, and *MC-DAGGES* showed worse accuracies than *fsANB-BDeu*, because the exact learning methods estimate the network structure more precisely than the greedy learned structure.

We compared *fsANB-BDeu* and *ANB-BDeu*. The difference between the two methods is whether the proposed PC search method is used. “Removed variables” in Table 4 represents the average number of variables removed from the class variable’s Markov blanket by our proposed PC search method. The results demonstrated that the accuracies of *fsANB-BDeu* tended to be much higher than those of *ANB-BDeu* when the value of “Removed variables” was large, such as Nos. 5, 12, 16, 34, and 38. Consequently, discarding numerous irrelevant variables in the features improved the classification accuracy.

Finally, we compared the runtimes of *fsANB-BDeu* and *GBN-BDeu* to demonstrate the efficiency of the ANB constraint. Table 8 presents the runtimes of *GBN-BDeu*, *fsANB-BDeu*, and the proposed PC search method. The results show that the runtimes of *fsANB-BDeu* were shorter than those of *GBN-BDeu* in all the datasets, because the execution speed of the exact learning ANB was almost twice that of the exact learning GBN, as described in Section 4. Moreover, the runtimes of *fsANB-BDeu* were much shorter than those of *GBN-BDeu* when our PC search method removed many variables, such as the No. 34 and No. 39 datasets. This is because the runtimes of *GBN-BDeu* decrease exponentially with the removal of variables, whereas our PC search method itself has a negligibly small runtime compared to those of the exact learning as shown in Table 8.

As a result, the proposed method *fsANB-BDeu* provides the best classification performances in all the methods with a lower computational cost than that of the *GBN-BDeu*.

## 7. Conclusions

First, this study compared the classification performances of the BNs exactly learned by BDeu as a generative model and those learned approximately by CLL as a discriminative model. Surprisingly, the results demonstrated that the performance of BNs achieved by maximizing ML was better than that of BNs achieved by maximizing CLL for large data. However, the results also showed that the classification accuracies of the BNs that learned exactly by BDeu were much worse than those that learned by the other methods when the class variable had numerous parents. To solve this problem, this study proposed an exact learning ANB by maximizing BDeu as a generative model. The proposed method asymptotically learned the optimal ANB, which is an I-map with the fewest parameters among all possible ANB structures. In addition, the proposed ANB is guaranteed to asymptotically estimate the identical conditional probability of the class variable to that of the exactly learned GBN. Based on these properties, the proposed method is effective for not only classification but also decision making, which requires a highly accurate probability estimate of the class variable. Furthermore, the learning ANB has lower computational costs than the learning BN does. The experimental results demonstrated that the proposed method significantly outperformed the approximately learned structure by maximizing CLL.

We plan on exploring the following in future work.

(1)It is known that neural networks are universal approximators, which means that they can approximate any functions to an arbitrary small error. However, Choi et al. [45] showed that the functions induced by BN queries are polynomials. To improve their queries to become universal approximators, they proposed a testing BN, which chooses a parameter value depending on a threshold instead of simply having a fixed parameter value. We will apply our proposed method to the testing BN.(2)Recent studies have developed methods for compiling BNCs into Boolean circuits that have the same input–output behavior [46,47]. We can explain and verify any BNCs by operating on their compiled circuits [47,48,49]. We will apply the compiling method to our proposed method.(3)Sugahara et al. [50] proposed the Bayesian network model averaging classifier with subbagging to improve the classification accuracy for small data. We will extend our proposed method to the model averaging classifier.

The above future works are expected to improve the classification accuracies and comprehensibility of our proposed method.

## Figures and Tables

**Figure 1 entropy-23-01703-f001:**
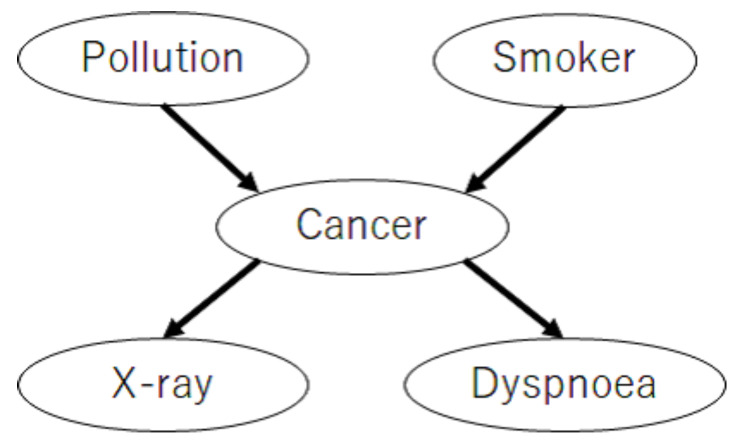
A network which satisfies Assumptions 2 and 3 (CANCER network [31]).

**Figure 2 entropy-23-01703-f002:**
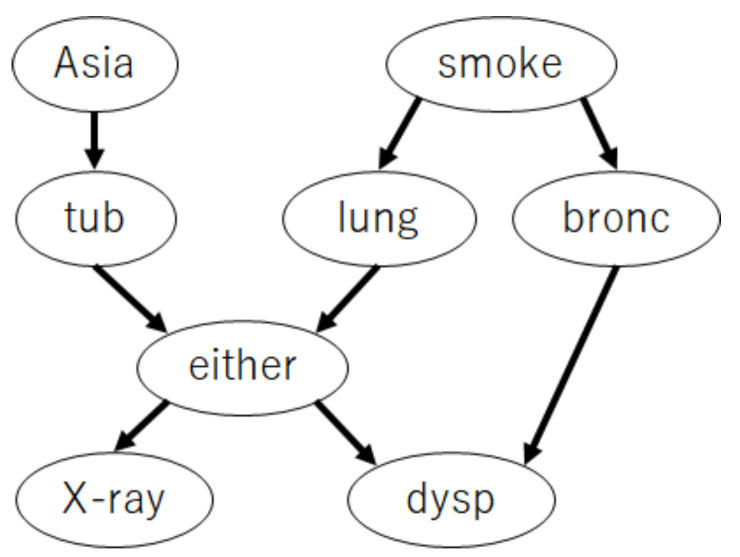
A network which violates Assumptions 2 and 3 (ASIA network [31]).

**Table 1 entropy-23-01703-t001:** Seven methods compared in the experiments.

Abbreviation	Methods
*Naive Bayes*	Navie Bayes classifier.
*GBN-BDeu*	Exact learning GBN method by maximizing BDeu.
*GBN-CMDL* [3]	Greedy learning GBN method using the hill-climbing search by minimizing CMDL while estimating parameters by maximizing LL.
*BNC2P* [3]	Greedy learning method with at most two parents per variable using the hill-climbing search by maximizing CLL while estimating parameters by maximizing LL.
*TAN-aCLL* [6]	Exact learning TAN method by maximizing aCLL.
*gGBN-BDeu*	Greedy learning GBN method using hill-climbing by maximizing BDeu.
*MC-DAGGES* [7]	Greedy learning method in the space of the Markov equivalent classes of MC-DAGs using the greedy equivalence search [19] by maximizing CLL while estimating parameters by maximizing LL.

**Table 2 entropy-23-01703-t002:** Computational environment.

CPU	2.2 GHz XEON 10-core processor
System Memory	128 GB
OS	Windows 10
Software	Java

**Table 3 entropy-23-01703-t003:** Classification accuracies of *GBN-BDeu*, *ANB-BDeu*, *fsANB-BDeu*, and traditional methods (bold text signifies the highest accuracy).

No.	Dataset	Variables	Sample Size	Classes	Naive Bayes	GBN-CMDL	BNC2P	TAN-aCLL	gGBN-BDeu	MC-DAGGES	GBN-BDeu	ANB-BDeu	fsANB-BDeu
1	Balance Scale	5	3	625	**0.9152**	0.3333	0.8560	0.8656	**0.9152**	0.7432	**0.9152**	**0.9152**	**0.9152**
2	banknote authentication	5	2	1372	0.8433	**0.8819**	0.8797	0.8761	**0.8819**	0.8768	0.8812	0.8812	0.8812
3	Hayes–Roth	5	3	132	0.8182	0.6136	0.6894	0.6742	0.7525	0.6970	0.6136	0.8182	**0.8333**
4	iris	5	3	150	0.7133	0.7800	0.8200	0.8200	0.8133	0.7800	**0.8267**	0.8200	0.8200
5	lenses	5	3	24	0.7500	0.8333	0.6667	0.7083	0.8333	0.8333	0.8333	0.7500	**0.8750**
6	Car Evaluation	7	4	1728	0.8571	**0.9497**	0.9416	0.9433	0.9416	0.9126	0.9416	0.9427	0.9416
7	liver	7	2	345	0.6319	0.6145	0.6290	**0.6609**	0.6029	0.6435	0.6087	0.6348	0.6377
8	MONK’s Problems	7	2	432	0.7500	**1.0000**	**1.0000**	**1.0000**	0.8449	**1.0000**	**1.0000**	**1.0000**	**1.0000**
9	mux6	7	2	64	0.5469	0.3750	0.5625	0.4688	0.4063	**0.7656**	0.4531	0.5469	0.5547
10	LED7	8	10	3200	0.7294	0.7366	**0.7375**	0.7350	0.7297	0.7331	0.7294	0.7294	0.7294
11	HTRU2	9	2	17,898	0.7031	0.7096	0.7070	0.7018	0.7188	0.7214	**0.7305**	0.7188	0.7161
12	Nursery	9	5	12,960	0.6782	**0.7126**	0.6092	0.5862	**0.7126**	0.6322	**0.7126**	0.6782	**0.7126**
13	pima	9	2	768	0.8966	0.9086	0.9118	0.9130	0.9092	0.9093	0.9112	**0.9141**	**0.9141**
14	post	9	3	87	0.9033	0.5823	**0.9442**	0.9177	0.9291	0.9046	0.9340	0.9181	0.9177
15	Breast Cancer	10	2	277	**0.9751**	0.8917	0.9473	0.9488	0.7058	0.6354	**0.9751**	**0.9751**	**0.9751**
16	Breast Cancer Wisconsin	10	2	683	0.7401	0.6209	0.6823	0.7184	0.7094	**0.9780**	0.7184	0.7040	0.7473
17	Contraceptive Method Choice	10	3	1473	0.4671	0.4501	**0.4745**	0.4705	0.4440	0.4576	0.4542	0.4650	0.4725
18	glass	10	6	214	0.5561	0.5654	0.5794	0.6308	0.4626	0.5888	0.5701	**0.6449**	0.5888
19	shuttle-small	10	6	5800	0.9384	0.9660	0.9703	0.9583	0.9683	0.9586	0.9693	**0.9716**	0.9695
20	threeOf9	10	2	512	0.8164	**0.9434**	0.8691	0.8828	0.8652	0.8750	0.8887	0.8730	0.8633
21	Tic-Tac-Toe	10	2	958	0.6921	**0.8841**	0.7338	0.7203	0.6754	0.7557	0.8340	0.8497	0.8570
22	MAGIC Gamma Telescope	11	2	19,020	0.7482	0.7849	0.7806	0.7631	0.7844	0.7781	0.7873	**0.7874**	0.7865
23	Solar Flare	11	9	1389	0.7811	0.8265	0.8315	0.8229	**0.8431**	0.8013	**0.8431**	0.8229	0.8373
24	heart	14	2	270	0.8259	0.8185	0.8037	0.8148	0.8222	**0.8333**	0.8259	0.8185	0.8296
25	wine	14	3	178	0.9270	**0.9438**	0.9157	0.9326	0.9045	**0.9438**	0.9270	0.9270	0.9270
26	cleve	14	2	296	0.8412	0.8209	0.8007	**0.8378**	0.7973	0.8041	0.7973	0.8277	0.8243
27	Australian	15	2	690	0.8290	0.8312	0.8348	0.8464	0.8420	0.8406	**0.8536**	0.8246	0.8522
28	crx	15	2	653	0.8377	0.8346	0.8208	0.8560	**0.8622**	0.8576	0.8591	0.8515	0.8591
29	EEG	15	2	14,980	0.5778	0.6787	0.6374	0.6125	0.6732	0.6182	0.6814	**0.6864**	**0.6864**
30	Congressional Voting Records	17	2	232	0.9095	0.9698	0.9612	0.9181	**0.9741**	0.9009	0.9655	0.9483	0.9397
31	zoo	17	5	101	0.9802	0.9109	0.9505	1.0000	0.9505	0.9802	0.9307	0.9505	0.9604
32	pendigits	17	10	10,992	0.8032	0.9062	0.8719	0.8700	0.9253	0.8359	**0.9290**	0.9279	0.9279
33	letter	17	26	20,000	0.4466	0.5796	0.5132	0.5093	0.5761	0.4664	0.5761	**0.5935**	0.5881
34	ClimateModel	19	2	540	0.9222	**0.9407**	0.9241	0.9333	0.9370	0.9296	0.9000	0.8426	0.9278
35	Image Segmentation	19	7	2310	0.7290	0.7918	0.7991	0.7407	0.8026	0.7476	0.8156	**0.8225**	**0.8225**
36	lymphography	19	4	148	**0.8446**	0.7939	0.7973	0.8311	0.7905	0.8041	0.7500	0.7770	0.7838
37	vehicle	19	4	846	0.4350	0.5910	0.5910	0.5816	0.5461	0.5414	0.5768	**0.6253**	0.6217
38	hepatitis	20	2	80	0.8500	0.7375	**0.8875**	0.8750	0.8500	**0.8875**	0.5875	0.6250	0.8375
39	German	21	2	1000	0.7430	0.6110	0.7340	**0.7470**	0.7140	0.7180	0.7210	0.7380	0.7410
40	bank	21	2	30,488	0.8544	0.8618	0.8928	0.8618	0.8952	0.8708	**0.8956**	0.8950	0.8953
41	waveform-21	22	3	5000	0.7886	0.7862	0.7754	0.7896	0.7698	0.7926	0.7846	0.7966	**0.7972**
42	Mushroom	22	2	5644	0.9957	**1.0000**	**1.0000**	0.9995	**1.0000**	0.9986	0.9949	**1.0000**	**1.0000**
43	spect	23	2	263	0.7940	0.7940	0.7903	0.8090	0.7603	0.8052	0.7378	**0.8240**	**0.8240**
	average				0.7764	0.7721	0.7936	0.7943	0.7867	0.7944	0.7963	0.8061	**0.8184**
	*p*-value (*ANB-BDeu* vs. the other methods)				0.00308	0.04136	0.00672	0.05614	0.06876	0.06010	0.22628	-	-
	*p*-value (*fsANB-BDeu* vs. the other methods)				0.00001	0.00014	0.00013	0.00280	0.00015	0.00212	0.00064	0.01101	-

**Table 4 entropy-23-01703-t004:** Statistical summary of *GBN-BDeu* and *fsANB-BDeu*.

No.	Variables	Classes	Sample Size	Parents	Children	Sparse Data	MB Size	Max Parents	Removed Variables
1	5	3	625	0.4	3.6	0.0	4.0	1.0	0.0
2	5	2	1372	0.0	2.0	0.0	4.0	4.0	0.0
3	5	3	132	3.0	0.0	17.2	3.0	1.0	1.0
4	5	3	150	1.8	1.2	0.0	3.0	2.0	0.0
5	5	3	24	1.1	1.0	0.0	2.1	1.1	2.0
6	7	4	1728	2.0	3.0	0.0	5.0	2.0	1.0
7	7	2	345	0.0	1.9	0.0	3.4	2.0	0.1
8	7	2	432	3.0	0.0	0.0	3.0	3.0	0.0
9	7	2	64	5.8	0.0	5.2	5.8	1.0	2.1
10	8	10	3200	0.9	6.1	0.0	7.0	1.0	0.0
11	9	2	17,898	1.8	4.2	0.0	4.2	2.0	0.9
12	9	5	12,960	4.0	3.0	0.0	0.0	0.0	8.0
13	9	2	768	1.4	1.7	0.0	7.0	4.0	0.0
14	9	3	87	0.0	0.0	0.0	7.0	3.0	0.1
15	10	2	277	0.9	8.0	0.0	1.0	1.0	0.0
16	10	2	683	0.7	0.3	0.0	8.9	2.0	5.0
17	10	3	1473	0.7	0.8	0.0	1.7	2.5	0.6
18	10	6	214	0.6	3.1	0.0	4.3	2.7	2.0
19	10	6	5800	2.0	4.0	0.0	7.0	5.0	1.9
20	10	2	512	5.0	2.1	0.0	7.6	2.7	0.2
21	10	2	958	1.2	2.2	0.0	5.3	3.0	0.3
22	11	2	19,020	0.0	6.1	0.0	8.0	4.0	1.7
23	11	9	1389	0.8	0.2	0.0	1.0	2.0	5.3
24	14	2	270	1.8	4.2	0.0	6.3	2.0	1.8
25	14	3	178	1.7	5.3	0.0	8.1	2.1	0.0
26	14	2	296	1.8	4.5	0.0	6.6	2.0	3.1
27	15	2	690	1.4	2.8	0.0	4.5	2.8	3.3
28	15	2	653	1.3	2.8	0.0	4.2	2.2	2.7
29	15	2	14,980	0.4	8.2	0.0	12.8	5.0	0.0
30	17	2	232	1.3	2.6	0.1	6.2	3.8	1.8
31	17	5	101	4.3	1.6	20.3	7.4	5.1	1.2
32	17	10	10,992	2.6	13.4	0.1	16.0	5.6	0.0
33	17	26	20,000	2.9	9.1	0.0	13.0	5.0	2.0
34	19	2	540	1.8	4.4	0.0	16.6	1.0	12.9
35	19	7	2310	0.7	10.4	0.0	13.2	4.0	0.0
36	19	4	148	1.6	5.9	0.2	13.1	2.2	5.3
37	19	4	846	1.1	5.1	0.1	10.1	4.1	0.5
38	20	2	80	1.3	6.1	0.4	16.0	6.9	5.4
39	21	2	1000	1.1	2.8	0.0	4.1	2.1	7.4
40	21	2	30,488	4.1	2.0	32.5	9.9	6.0	4.0
41	22	3	5000	3.8	10.1	0.0	14.5	4.0	2.0
42	22	2	5644	1.3	3.3	9.0	6.4	6.4	0.0
43	23	2	263	2.0	3.4	0.0	7.7	3.0	0.0

**Table 5 entropy-23-01703-t005:** The SHD between the structure learned by the proposed method and the I-map with the fewest parameters among all the ANB structures and the KLD between the learned class variable posterior by the proposed method and learned one using the true structure.

Network	Variables	Sample Size	SHD-(Proposal, I-Map ANB)	KLD-(Proposal, True Structure)
		100	3	$2.31\times 10^{-2}$
		500	2	$1.24\times 10^{-1}$
		1000	2	$7.63\times 10^{-2}$
ASIA	8	5000	1	$3.67\times 10^{-3}$
		10,000	0	$9.26\times 10^{-4}$
		50,000	0	$6.28\times 10^{-4}$
		100,000	0	$3.59\times 10^{-5}$
		100	1	$8.79\times 10^{-2}$
		500	1	$2.43\times 10^{-3}$
		1000	0	0.00
CANCER	5	5000	0	0.00
		10,000	0	0.00
		50,000	0	0.00
		100,000	0	0.00

**Table 6 entropy-23-01703-t006:** Missing variables, extra variables, and runtimes (ms) of each method.

Network	Variables	MMPC			HITON-PC			PCMB			$S^{2}$ TMB			Proposal		
		Missing	Extra	Runtime	Missing	Extra	Runtime	Missing	Extra	Runtime	Missing	Extra	Runtime	Missing	Extra	Runtime
ASIA	8	1.25	0.00	251	1.75	0.63	117	1.75	0.63	163	0.25	0.50	888	0.00	3.50	13
SACHS	11	1.91	0.00	1062	2.64	0.36	248	2.00	0.00	610	0.00	0.00	4842	0.00	2.55	12
CHILD	20	1.75	0.05	6756	2.35	0.95	380	2.00	0.25	1191	0.05	0.05	6669	0.00	11.80	16
WATER	32	3.59	0.00	407	4.00	0.19	140	3.78	0.31	260	2.03	1.47	29,527	0.25	13.44	25
ALARM	37	1.81	0.14	3832	2.38	0.57	281	2.19	0.19	1025	0.14	0.11	11,272	0.05	10.92	39
BARLEY	48	2.85	1.23	4928	3.46	0.42	269	3.19	0.42	830	1.15	0.46	99,290	0.38	9.75	49
Average		2.19	0.24	2872	2.76	0.52	239	2.48	0.30	680	0.60	0.43	25,415	0.11	8.66	26

**Table 7 entropy-23-01703-t007:** Average classification accuracy of each method.

	MMPC	HITON-PC	PCMB	$S^{2}$ TMB	Proposal
Average	0.6185	0.6219	0.6302	0.7980	0.8164

**Table 8 entropy-23-01703-t008:** Runtimes (ms) of GBN-BDeu, fsANB-BDeu, and the proposed PC search method.

No.	Variables	Sample Size	Classes	GBN-BDeu	fsANB-BDeu	The Proposed PC Search Method
1	5	625	3	169.4	23.0	6.3
2	5	1372	2	19.3	10.3	2.0
3	5	132	3	15.6	3.0	0.2
4	5	150	3	16.7	5.0	0.2
5	5	24	3	15.3	1.0	0.1
6	7	1728	4	90.8	22.9	1.7
7	7	345	2	21.1	15.6	0.3
8	7	432	2	31.0	20.7	0.5
9	7	64	2	18.9	9.1	0.1
10	8	3200	10	114.6	55.1	3.1
11	9	17,898	2	300.5	251.3	10.2
12	9	12,960	3	707.4	525.8	5.8
13	9	768	9	66.8	27.6	0.6
14	9	87	5	39.6	0.3	0.1
15	10	277	2	162.6	6.9	0.3
16	10	683	2	453.1	258.9	0.4
17	10	1473	3	161.1	121.4	0.8
18	10	214	6	63.0	22.3	0.2
19	10	5800	6	159.6	67.2	2.8
20	10	512	2	102.7	58.2	0.4
21	10	958	2	212.2	193.0	0.5
22	11	19,020	2	979.8	277.2	5.3
23	11	1389	9	379.4	17.2	0.9
24	14	270	2	1988.6	299.8	0.1
25	14	178	3	1233.7	585.0	0.1
26	14	296	2	2034.5	115.2	0.2
27	15	690	2	10,700.3	927.6	0.3
28	15	653	2	23,069.5	2774.3	0.2
29	15	14,980	2	12,407.6	8248.8	4.1
30	17	232	2	11,682.6	1623.6	0.2
31	17	101	5	7326.5	1985.1	0.1
32	17	10,992	10	84,967.1	48,636.9	3.4
33	17	20,000	26	339,910.2	30,224.8	6.3
34	19	540	2	217,457.0	12.0	0.3
35	19	2310	7	190,895.9	103,447.5	1.0
36	19	148	4	107,641.8	1171.4	0.2
37	19	846	4	144,669.5	62,663.0	0.4
38	20	80	2	98,841.9	821.6	0.1
39	21	1000	2	2,706,616.6	8885.1	0.5
40	21	30,488	2	1,562,6734.5	130,491.6	11.8
41	22	5000	3	10,022,030.7	757,611.7	2.1
42	22	5644	2	4,640,293.5	2,382,657.7	2.3
43	23	263	2	2,553,290.4	1,386,088.2	0.2

## Data Availability

The datasets used in our experiments are available at: http://www.ai.lab.uec.ac.jp/software/ (accessed on 29 November 2021).

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
