# Peer review of "Exact Learning Augmented Naive Bayes Classifier"

_entropy, 2021, doi:10.3390/e23121703_

Round 1

Reviewer 1 Report

Interesting paper. I am reading it carefully. A little too long and takes really time to read. It is good for small number of readers, the specialists.

When I accepted to review, I requested more time, but finally, I could do it faster.

  • To understand the introduction and position of the paper, it is necessary to explain a little the methods with which this paper is compared. This done in details in the Background section, but the reader may abandon reading before that.
  • A little more explanation on the significance, choice and sensitivity of the results on the hyperparameters will be good.
  • Give more expressive captions for figures and tables. In particular Figures 1 and 2.
  • There are too many conclusions in the discussion section. At the end, the reader does not get easily which method is good. A more synthetic conclusions diagram may help the reader.

Reviewer 2 Report

This paper compares the classification accuracies of BNs with approximate learning using CLL to those with exact 7 learning using ML. and the authors propose an exact learning augmented naive Bayes classifier, which guarante to asymptotically estimate the identical class posterior to that of the exactly learned BN and demonstrate a superior performance.

The methods are adequately described, although some elements appear in the notation without description, which slightly differs the reading of the work.

Regarding the results, presenting them in another order would help the reader to understand them better and form their own conclusions. Your description should appear in the first table and the results of the work in the following tables.

Reviewer 3 Report

Review of "Exact Learning Augmented Naive Bayes Classifier" by Sugahara and Ueno (2021)

Authors proposed an augmented Naive bayes classifier method to improve exact learning Bayesian Networks using the Marginal Likelihood. The work is well presented and results are auspicious. I have some comments/suggestions to help to authors to improve tha manuscript:

1. Merge paragraphs in L40-41, 53-54, 80-81, 84-85, 187-188, 189-190

2. L126: "Verma and Pearl [19]".

3. L127: "(Verma and Pearl [19])".

4. Before Eq. (4): "(MDL) score presented in (4), which" and delete "presented below".

5. L559: Authors could include in further work list that other procedures such as Metropolis-Hastings algorithm could be employeed when the large data set include extreme observations that produced heavy-tailed disitributions, such as Student-t one (Contreras-Reyes et al., 2018).

Reference:

Contreras-Reyes, J.E., Quintero, F.O.L., Wiff, R. (2018). Bayesian modeling of individual growth variability using back-calculation: Application to pink cusk-eel (Genypterus blacodes) off Chile. Ecological Modelling 385, 145-153.
